# Willingness of advanced cancer patients to receive palliative care and its determinants: A cross-sectional study in Northern Tanzania

**Grace Leonard Mushi**[1,2]*, **Furaha Serventi**[3], **Julius Pius Alloyce**[3], **Vivian Frank Saria**[2], **Xianghua Xu**[1], **Khalid Khan**[1], **Qinqin Cheng**[1], **Yongyi Chen**[1,4]

**1** Xiangya School of Nursing, Central South University, Changsha City, Hunan Province, People's Republic of China, **2** Department of Nursing, Kilimanjaro Christian Medical Centre, Moshi, Kilimanjaro, Tanzania, **3** Department of Oncology, Kilimanjaro Christian Medical Centre, Moshi, Kilimanjaro, Tanzania, **4** Nursing Department, Hunan Cancer Hospital and The Affiliated Cancer Hospital of Xiangya School of Medicine, Changsha, Hunan Province, China

* gracemushy2@gmail.com

**Data Availability Statement:** All relevant data are within the paper.

## Abstract

### Background

The purpose of this study was to assess willingness of advanced cancer patients to receive palliative care and its determinants at Cancer Care Centre in Kilimanjaro Christian Medical Centre Northern Tanzania.

### Objective

The purpose of this study was to assess willingness of advanced cancer patients to receive palliative care and its determinants at Cancer Care Centre in Kilimanjaro Christian Medical Centre Northern Tanzania.

### Methods

This was an institution-based cross-sectional study and the target population was all advanced cancer patients attending care at Cancer care Centre in Northern Tanzania. Data was collected using a structured questionnaire and analysed using Stata for windows 15. A descriptive analysis was conducted to summarize the data using mean standard deviation, while categorical data was summarized using frequency and percentages. Both bivariate and multivariate logistic regression analysis was used to determine the predictors associated with willingness to receive palliative care.

### Results

The results showed that more than half of the respondents had willingness to accept palliative care. The degree of willingness PC among advanced cancer patients was high 60.6% (95%CI: 55.8–65.3). The predictors which remained significant associated with willingness to receive palliative care in multivariate analysis were the knowledgeable of palliative care

**Funding:** The authors received no specific funding for this work.

**Competing interests:** The authors have declared that no competing interests exist.

[AOR: 2.86; 95%CI: 1.69–4.85] and high perceived benefits of palliative care [AOR: 3.58; 95%CI: 2.12–6.04].

## Conclusion

Willingness to accept palliative care services was more than half of the patients just 60.6% among patients with advanced cancer from the study site. Advanced age of a patient, occupations, better knowledge, and perceived benefits for palliative care was the significant predictor for patients' willingness to accept palliative care.

## Introduction

Cancer is already one of the foremost causes of death worldwide. It is a major developing public health fear in Sub-Saharan Africa (SSA); most patients present late when the disease is in advanced stages when chemotherapy and other modalities of treatment may not offer additional therapeutic benefit [1]. However, access to cancer screening and essential treatment services is inadequate in many places of SSA. Palliative care remains an urgently neglected need in developing world, especially in advanced cancer patients [1, 2]. Alteration from curative to palliative care might be one of the most decisions that people and their families face in the late stages of advances illness [3]. Still, Palliative care is typically being delivered very late in the progression of the disease when medication therapy is disturbed due to the absence of new options and the end of life due to difficulty in controlling the symptoms. It is more effective if it is integrated early into standard oncology care when advanced terminal cancer is first diagnosed. But many advanced cancer patients still die with no access to palliative care [4]. It is about options for care at the end of life (EoL) that can be influenced by differences inpatient and their caregiver, fluctuating perceptions of the disease and prognosis, which increase the need for care, communication difficulties, and family conflict [3]. The role of multidisciplinary treatment planning in which each contributes to the interpretation of diagnostic and staging information is now regarded as best standard practice [5].

The figure of patients identified with cancer is progressively rising, and this is due to the development of cancer detection and treatment [6]. The changeover from curative to Palliative care can be one of the most challenging decisions people and their families face in the later stages of an advanced illness [3]. The benefits of including PC in oncology care are known and ideal timing for cancer patients' referral to these services. Moreover, the existence of oncological referral centers and public health rules which recognize PC as an essential component in health care practice, the reality has not yet been integrated in clinical practice [4]. Concerning patients who have advanced cancer, many improvements have already been attempted and implemented in the form of multidisciplinary meetings, case conferences, psychosocial support, and tumor board decision. There are still barriers regarding communication about prognosis, EoL issues, and palliative care [7].

Palliative care has a positive effect on symptom burden, quality of life, psychosocial communication, prognostic understanding and decision making, mood, satisfaction with the care received, and quality of care at the end of life. It is more effective if it is integrated early into standard oncology care when advanced terminal cancer is first diagnosed [4, 8]. Patient's attitudes or beliefs about particular treatments are often predictive of their future health behaviors. Likewise, educational achievement is another crucial determinant of EoL attitudes and health behaviors. Patient with fewer years of education are likely to believe that their incurable cancer can be cured or more likely to receive ineffective, troublesome treatment at EoL [9]. Most people with cancer present to the health facilities with incurable disease. Disease-

modifying treatment is not available to most patients, PC offers precise public health approach to cancer care in this setting [10].

Sufferers with advanced most cancers require a range of offerings to make assured their physical, psychological, social and spiritual desires are encountered successfully and to facilitate them to live and die in the place of their choices, if at all possible [8]. However, PC is nowadays acknowledged EoL care that brings various benefits, some benefits include increased care quality, an opportunity for patients to have a peaceful and honorable death, better quality of dying, family caregiver's advance satisfaction, and bereavement adjustment. Therefore the decision to receive palliative care is always made by family members when patients near end of their lives but willingness to receive palliative care still remain unclear [11].

Studies have mainly focused on the scope of symptoms and palliative care needs of cancer patients in hospice and palliative care, but some have focused on cancer patients in hospital settings. However, we still knowledge regarding the extent of palliative care needs and incidence and symptoms in acute care settings in places with limited availability of specialized palliative care [12]. Early recognition of the widespread requirements of cancer patients is essential not only to ease the suffering initiated to the patient but also to improve the quality of cancer care. So it very important to understand the broad range needs of cancer patients for developing and enhancing services to address the identified gaps in cancer care [13]. Despite the known benefits of palliative and hospice care on enhancing patient's quality of life these services remain underutilized.

Hence, this study aimed to investigate the willingness of advanced cancer patients to receive palliative care (PC), to explore the association between willingness to accept PC, and investigate the association between willingness to receive PC, knowledge and attitude of the patients.

## Material and methods

### Study design and settings

This was a cross-sectional hospital-based study which was conducted between October 2020 and November 2020. The research was conducted at Kilimanjaro Christian Medical Center (KCMC), referral hospital in Northern Tanzania. The catchment area for KCMC includes the five regions in the northern part of Tanzania namely Kilimanjaro, Arusha, Manyara, Singida and Tanga with a population of about 15 million people. The hospital offers general and specialized medical services including oncology. It is a busy tertiary referral hospital which receives patients with chronic illnesses of which cancer and HIV/AIDS is the most common conditions potentially requiring palliative care in this setting.

### Study population and sample size

The target group was advanced cancer patients who were attending KCMC Cancer Care Centre (CCC). A convenient sampling method was used to obtain participants for the study from KCMC Cancer Care Centre (CCC). The inclusion criteria were patients with a confirmed diagnosis of advanced cancer, being aware of diagnosis and prognosis, reading and filling in the questionnaire, age 18 years or older, and being able to communicate in Swahili or English. We excluded new patients who were being seen for the first time, and unconscious cancer patients.

We determined the sample size for cancer patients who filled the questionnaires by applying the Cochran's formula: sample size, $n = (\text{Z-score})^2 * p * (1-p) / (\text{margin of error})^2$ where, Z-score = 1.96, p = 0.5 (p is the estimated proportion of compliance in the population), and margin of error = 5% [14]. This gives a minimum sample of 384, with an addition of 10% for

non-response rate resulted to a desired sample 423 patients. A 25 patients were excluded and thus, 398 patients were included in the final analysis.

## Data collection method and tools

Data was collected through a structured questionnaire. An interview schedule method was employed to obtain the data from the participants whereby the two trained nurses (research assistants) were using structured questionnaire to assess the respondents and participants fill in the questionnaire. The questionnaire was adopted from Chen et al., 2019. This instrument was initially developed and validated by Wen Yu Hu in Taiwan. It was in Chinese version and it has been translated to English and then Swahili by two bilingual specialists to assess the comprehensive willingness of advanced cancer patients in Tanzania. This is because data was collected using Swahili the spoken language of participants. The tool consisted of four sections, namely socio-demographic and clinical characteristics, willingness to receive palliative care, knowledge about palliative care, attitude toward palliative care.

## Study variables

**Socio-demographic and medical variables.** The questionnaire composed of 16 items regarding socio-demographic and medical variables which was designed to collect demographic information from participants such as age, gender, educational level, marital status, occupation, place of residence, religious situation, financial situation, diagnosis, duration proven to have cancer and treatment received was retrieved from hospital information systems at the participant center, if discussed about end of life treatment preferred and whom have the patient discussed with about it, currently ward which has been admitted has been removed because the center is for outpatient clinic. Their questionnaire has been structured and validated thus, includes socio demographic information in series questions, EoL care preference, scale for measuring the patient's willingness to receive palliative care, knowledge and attitude towards palliative care. The statistical description of the socio-demographics and clinical characteristics was carried out by frequency tables, means, and standard deviations. Each measurement scale was constructed after discovering face validity, content validity based on expert's rating and factor analysis. All the scale would remain to initiate good reliability and validity.

**Willingness to PC.** The patients' willingness was scored based on 5-point Likert scale ranging from 'very unwilling' (score 1) to 'very wiling' (score 5). Higher total score shows greater willingness to receive palliative care. The cronbach's α for this scale has 0.74.

**Knowledge of PC.** The questionnaire has 12 items; true/false/I don't know questions with each accurate answer being scored as 1 and each incorrect answer or don't know scored 0, total score range from 0–12. Higher score indicates better knowledge about pc. This scale was established for content validity based on experts' ideas, consistency, item difficulty, discrimination, and test-retest reliability. Cronbach's α was done to identify the point to which it was error-free among 15 cancer patients. The results showed Cronbach's α score of 0.82 which indicate high internal consistency. The items with less internal consistency were deleted and adjustment was made for reliability of the study tool.

*Attitude of PC.* The questionnaire has 20 items that assess attitude towards PC. Scale includes two subscales' perception of the advantages of PC' and 'perception of the disadvantages of PC. A5 point likert's scale was adopted to measure the level of agreement from 'strongly disagree' (I point) to 'strongly agree' (5 points). Higher score shows a more positive attitude towards PC. In expressions of internal consistency, this scale has Cronbach's α of 0.85.

## Data processing and statistical analysis

Data analysis was performed using Stata 15. Numerical data were summarized using mean and standard deviation, while categorical variables were summarized using frequency and percentage. The proportion for willingness was obtained from the willingness score from the likert scale with an average (score>3 to the maximum score 5) being used as the cup-point for willingness otherwise was regarded as not willing to receive PC.

Further analysis using logistic regression analysis was conducted to determine the determinants for the willingness to receive palliative care for which the bivariate logistic regression analysis was done for the crude odds ratio (COR) to determine the independent predictors associated with Noncompliance. From the bivariate analysis, variables with p value less than 0.2 was included in multivariate logistic regression analysis and adjusted odds ratio (AOR) were obtained at 95% confidence interval; a multivariate analysis was conducted to control for the possible confounders. Variable with 95%CI that does not include null value 1 was regarded to be statistically significant determinants for the willingness to receive PC. Findings were presented using tables and figures.

## Ethical considerations

Formal approval was obtained from the Ethics and Research Committee of Xiangya School of Nursing, Central South University (**approval number E2020118**). With this clearance, a formal approval was sought from the KCMC Hospital Administration, the Head of Department of Oncology (Cancer Care Centre). This study posed no potential risks but may rather yield benefits to the KCMC hospital. Participants were asked for their informed consent before data collection and we included only those who provided their informed consent. Privacy and confidentiality of information recorded on all clients recruited in this study was ensured, no names of the clients were recorded instead study ID was used as unique identifier. Participants' freedom of giving their responses and the opportunity to withdrawal from the study at any time they wished. All information from the field was secured and used for the purpose of this research and not otherwise.

## Results

### Schematic presentation of study participants

A total of 423 cancer patients attending care at KCMC Cancer Care Centre were asked for their participation in this study. Twenty five (25) of patients did not meet the inclusion criteria and were excluded from the analysis (4 refused to participate, 16 very sick patients and 5 were missing potential variables/incomplete). Thus, a total 398 were included in the final analysis. Of 398 eligible participants 241(60.6%) had willingness to receive PC and 157(39.4%) had no willingness to receive PC (Fig 1).

### Socio-demographic and economic status

Of 398 patients recruited in this study, about half 202(50.8%) were female and 196(49.2%) were male. The mean age of the participants was 55 years old with the standard deviation of 16.3 years. More than half 208(52.3%) were older aged 55 years and above. About three quarter 302(75.9%) were married. Only 59(14.8%) reached college or university while majority 189 (47.5%) were had primary education and 183(46%) were farmers. About 222(55.8%) were residing in rural areas. The study site is dominated by Christians 329(82.7%) (Table 1).

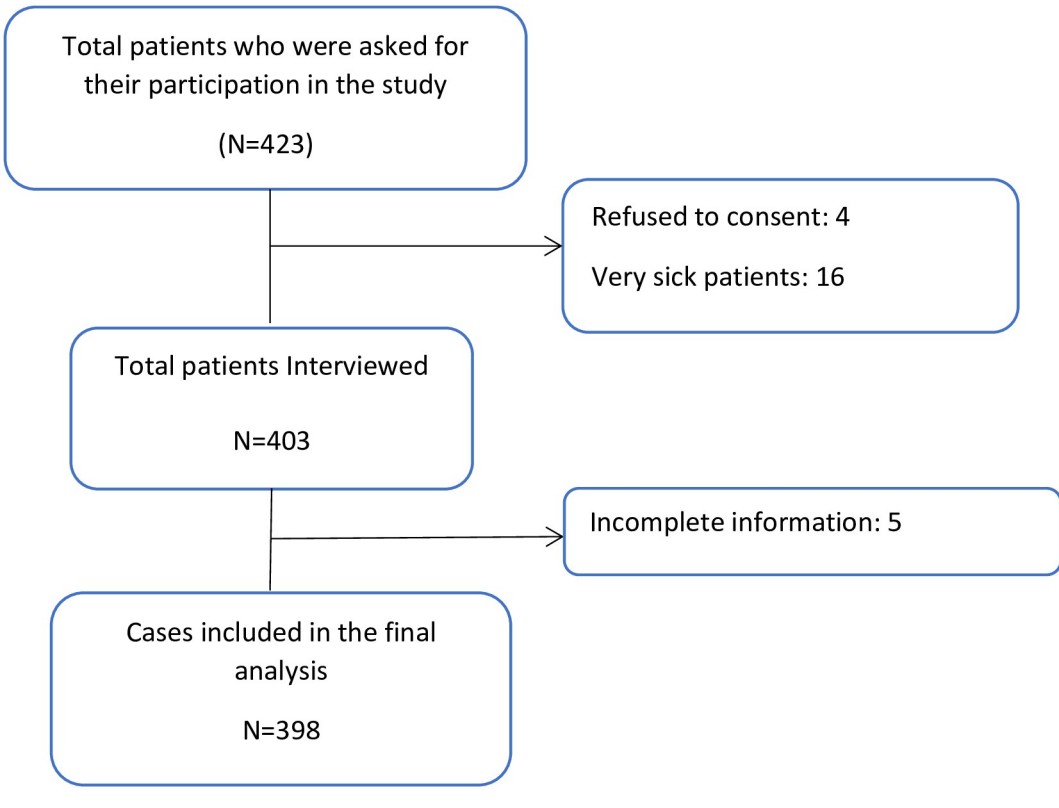

**Fig 1. Schematic presentation of the participants.**

**Type of cancers.** The most common cancers among the study respondents included reproductive cancer 113(28.4%), genitourinary cancers 77(19.3%), lymphatic system cancers 77(19.3%) and digestive cancers 63(15.8%) (Fig 2).

## The willingness to accept palliative care

The degree of accepting PC among advanced cancer patients was high 60.6% (95%CI: 55.8–65.3). The most palliation with highest demand among the studied participants includes tracheal intubation, artificial ventilation, artificial nutrition and hydration, tube feeding and palliative medicines. Majority 75.4% had willingness of receiving palliative care at home as compared to 67.8% who had also showed their willingness to get palliation under-hospital management (Table 2).

## Determinants for the willingness to accept PC

Table 3 shows the determinants for the willingness to receive PC. Variables which showed statistically significant positive association with willingness to receive PC among the studied participants were knowledgeable of palliative care [COR: 2.08; 95%CI: 1.36–3.16] and the perceived benefits of PC [COR: 3.00; 95%CI: 1.94–4.64]. We significant negative association with formally employed [COR: 0.48; 95%CI: 0.25–0.91]; having public medical insurance as compared to own expenses [COR = 0.37; 95%CI: 0.18–0.78] and longer duration living with cancer more than 6 months [COR:0.46; 95%CI: 0.22–0.96]. In multivariate logistic regression analysis, determinants which remained significant associated with willingness to receive

**Table 1. Socio-demographic and economic status (N = 398).**

| Variables | N | % |
|---|---|---|
| **Age (years)** | | |
| <45 | 101 | 25.4 |
| 45–54 | 89 | 22.4 |
| 55+ | 208 | 52.3 |
| [Mean; SD] | [55; 16.3] | |
| **Sex** | | |
| Male | 196 | 49.2 |
| Female | 202 | 50.8 |
| **Marital status** | | |
| Single | 29 | 7.3 |
| Married | 302 | 75.9 |
| Widow, separated, divorced | 67 | 16.8 |
| **Education status** | | |
| Never been to school | 32 | 8.0 |
| Primary school | 189 | 47.5 |
| Secondary school | 118 | 29.6 |
| College or University | 59 | 14.8 |
| **Occupation** | | |
| Farmer | 183 | 46.0 |
| Informal employed | 79 | 19.8 |
| Formal employed | 51 | 12.8 |
| Other (retired, jobless) | 85 | 21.4 |
| **Place of residence** | | |
| Urban | 176 | 44.2 |
| Rural | 222 | 55.8 |
| **Religion** | | |
| Christian | 329 | 82.7 |
| Muslim | 67 | 16.8 |
| No religion | 2 | 0.5 |

palliative care includes knowledgeable of PC [AOR: 2.86; 95%CI: 1.69–4.85] and high perceived benefits of PC [AOR: 3.58; 95%CI: 2.12–6.04]. Advanced age, female patients, normal/high economic status, and having positive attitudes on PC showed positive association for the willingness to receive PC, but these were not statistically significant and thus, these association could be explained by other factors (Table 3).

## Obstructing barriers to accept palliative care among advanced cancer patients

Different obstructing barriers to accept PC was observed among cancer patients; these includes (i) barriers to be informed by PC team about terminal illness including low willingness to be informed, poor medical knowledge, patient survival, patient condition and poor support for the patients; (ii) poor understanding of patient on PC offers, alternative treatment, poor emotional stability; (iii) palliative care team obstacles such as low willingness of family members to inform patients of their terminal conditions, poor response to curative treatment, poor family support for patients, worried about being abandoned by the paramedics, Palliative care costs cannot be covered by personal insurance (Table 4).

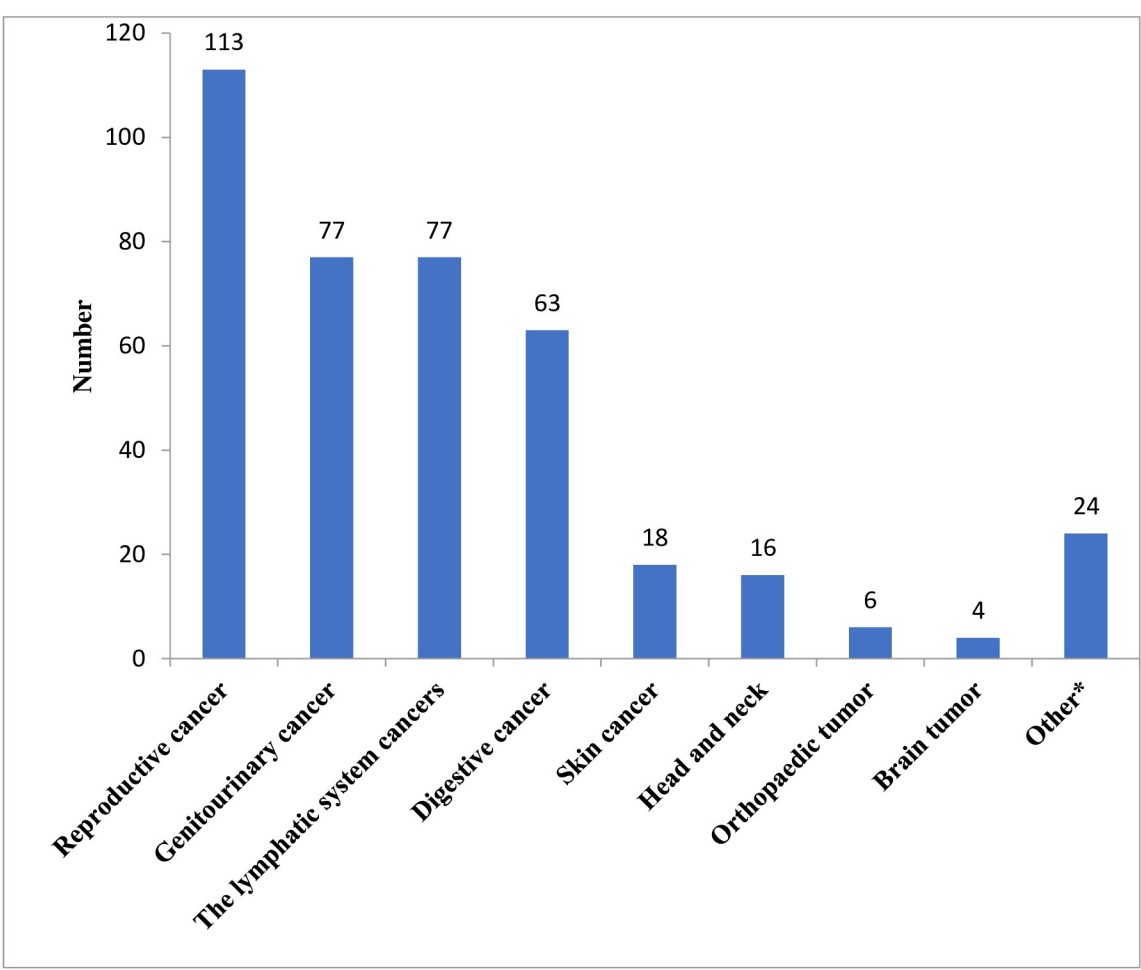

**Fig 2. Distribution of type of cancers among respondents.** *Multiple myeloma, lung cancer.

**Table 2. The willingness to receive palliative care (n = 398).**

| Willingness items (multiple responses) | Number | Percentage | 95%CI for the percentage |
|---|---|---|---|
| Tracheal intubation | 324 | 81.4 | 77.6–85.2 |
| Artificial ventilation | 275 | 69.1 | 64.3–73.9 |
| Cardiopulmonary resuscitation (CPR) | 397 | 99.7 | 99.2–100 |
| Artificial nutrition and hydration | 333 | 83.7 | 80.2–87.2 |
| The tube feeding | 327 | 82.2 | 78.4–85.7 |
| Palliative medicines | 278 | 69.8 | 65.3–74.4 |
| **Place of preference to receive PC** | | | |
| Under hospital management | 338 | 84.9 | 81.7–88.7 |
| Home management | 300 | 75.4 | 71.4–79.6 |
| Under any health care facility | 270 | 67.8 | 63.6–72.1 |
| An overall willingness to receive PC | 241 | 60.6 | 55.8–65.6 |

**Table 3. Determinants for the willingness to receive PC among advanced cancer patients (N = 398).**

| Variables | Willingness | | Crude OR(95%) | Adjusted OR(95%) |
|---|---|---|---|---|
| | **Low** | **High** | | |
| **Age (years)** | | | | |
| <45 | 47(46.5) | 54(53.5) | ref | |
| 45–54 | 35(39.3) | 54(60.7) | 1.34(0.75–2.40) | 1.58(0.81–3.09) |
| 55+ | 75(36.1) | 133(63.9) | 1.54(0.95–2.51) | 1.39(0.75–2.59) |
| **Gender** | | | | |
| Male | 76(38.8) | 120(61.2) | ref | |
| Female | 81(40.1) | 121(59.9) | 0.95(0.63–1.42) | 1.03(0.64–1.66) |
| **Marital status** | | | | |
| Single | 12(41.4) | 17(58.6) | ref | |
| Married | 117(38.7) | 185(61.3) | 1.12(0.51–2.42) | 0.97(0.39–2.41) |
| Other* | 28(41.8) | 39(58.2) | 0.98(0.40–2.39) | 0.94(0.33–2.69) |
| **Occupation** | | | | |
| Farmer | 61(33.3) | 122(66.7) | ref | |
| Informal employed | 34(43.0) | 45(57.0) | 0.66(0.38–1.14) | 0.59(0.32–1.10) |
| Formal employed | 26(51.0) | 25(49.0) | 0.48(0.25–0.91) | 0.68(0.31–1.48) |
| Other (retired, jobless) | 36(42.4) | 49(57.6) | 0.68(0.40–1.16) | 0.70(0.38–1.29) |
| **Medical expenses** | | | | |
| Own expenses | 46(33.6) | 91(66.4) | ref | |
| Public expenses | 23(57.5) | 17(42.5) | 0.37(0.18–0.78) | 0.46(0.21–1.01) |
| Medical insurance | 88(39.8) | 133(60.2) | 0.76(0.49–1.19) | 0.74(0.43–1.25) |
| **The duration proven you have cancer** | | | | |
| 1-3months | 11(25.0) | 33(75.0) | ref | |
| 4-6months | 21(38.2) | 34(61.8) | 0.54(0.22–1.31) | 0.81(0.31–2.15) |
| > 6months | 125(41.8) | 174(58.2) | 0.46(0.22–0.96) | 0.51(0.24–1.12) |
| **Perceived current health status** | | | | |
| Poor | 94(36.7) | 162(63.3) | ref | |
| Good | 63(44.4) | 79(55.6) | 0.73(0.48–1.11) | 0.77(0.44–1.36) |
| **Economic status** | | | | |
| Low | 63(44.7) | 78(55.3) | ref | |
| Normal | 94(36.6) | 163(63.4) | 1.40(0.92–2.13) | 1.40(0.85–2.29) |
| **Discussed End life medications** | | | | |
| No | 46(43.4) | 60(56.6) | ref | |
| Yes | 111(38.0) | 181(62.0) | 1.25(0.80–1.96) | |
| **Whom did you discuss with** | | | | |
| Family member/friend | 81(36.8) | 139(63.2) | ref | |
| Health care provider | 30(41.7) | 42(58.3) | 0.82(0.47–1.40) | |
| **Knowledge on PC** | | | | |
| Not knowledgeable | 101(47.4) | 112(52.6) | ref | |
| Knowledgeable | 56(30.3) | 129(69.7) | 2.08(1.36–3.16) | 2.86(1.69–4.85) |
| **Attitude of PC** | | | | |
| Negative | 72(36.0) | 128(64.0) | ref | |
| Positive | 85(42.9) | 113(57.1) | 0.75(0.50–1.12) | 1.12(0.67–1.89) |
| Perceived benefits of PC | | | | |
| Low | 105(52.0) | 97(48.0) | ref | |
| High | 52(26.5) | 144(73.5) | 3.00(1.94–4.64) | 3.58(2.12–6.04) |

* Widow, separated, divorced

**Table 4. Obstructing barriers to accept palliative care among advanced cancer patients.**

| Variables | Number | % |
|---|---|---|
| **Can hinder me to be informed by palliative care team about terminal illness** | | |
| Low willingness to be informed of terminal illness | 256 | 64.3 |
| Poor medical knowledge | 166 | 41.7 |
| Patient survival is short | 260 | 65.3 |
| Patient condition worsen rapidly | 278 | 69.8 |
| Poor family support for the patients | 253 | 63.6 |
| **Will prevent me from signing the blocker does not accept CPR** | | |
| Poor understanding of the patient | 278 | 69.8 |
| Alternative medical treatment | 246 | 61.8 |
| Lack of trust relationship between family members and patients | 257 | 64.6 |
| Poor emotional stability of the patient | 284 | 71.4 |
| I don't know what palliative care offers | 308 | 77.4 |
| **Will hinder my acceptance of palliative care team obstacles** | | |
| Low willingness of family members to inform patients of their terminal conditions | 286 | 71.9 |
| Poor response to curative treatment | 209 | 52.5 |
| Poor family support for patients | 248 | 62.3 |
| I'm worried about being abandoned by the paramedics | 224 | 56.3 |
| Palliative care costs cannot be covered by personal insurance | 272 | 68.3 |

## Discussion

This study included three hundred ninety-eight patients with advanced cancer attending care at Cancer Care Centre in Northern Tanzania; the majorities were older, aged at least 55 years. The results showed high percent 60.6% of the participants had willingness to receive palliative care. Tracheal intubation, artificial ventilation, artificial nutrition and hydration, tube feeding and palliative medicines were highly recognized by the participants.

Being knowledgeable and high perceived benefits of palliative care were the most significant determinants for the willingness of PC. Similarly with the recent study in Taiwan by [11] which found that the knowledge about PC in hospital was the significant predictor for patients' willingness to accept PC. Knowledge plays a big role in increasing understanding the importance of PC for the advanced disease or non-curative disorders; this was also observed in Spanish. The study conducted by [15] found that sufficient knowledge on palliative care was significant factor for its acceptance [15]. Another study in Mexico border region of Southern California by [16] and others in 2017 showed that knowledge on services provided was the factor for acceptance and needs of PC [16] Similarly to the study in Taiwan and China by [6] which reported similar idea; however also indicated that it is very important to understand the comprehensive needs of cancer patients for improving palliative care service [6]. This implies that, the needs would be the attributed barriers for knowledge on PC. The majority of patients with advanced cancer lose hope and some badly affected socially, economically and psychologically that needs much social and psychological help which is among PC services.

The attitude score on PC significant factor for the decrease in willingness score to accept PC. Inconsistency with the study in the United States of America, reported that misinformation and misperceptions were critical to further acceptance of this care [13]. The findings from the current study are different from the study by [11] in Taiwan, which found that a positive attitude was the significant predictor of willingness to receive PC services. The difference can be explained due to the nature of the study; for example, the research in Taiwan was centered

on hospice palliative care, while the current study focused on all palliative care treatment, whether in hospitals or home PC.

The findings from the current study also found different obstructing barriers to accepting palliative care was observed among cancer patients; these include obstacles to be informed by PC team about terminal illness, poor understanding of patient on PC offers, the low willingness of family members to educate patients on their terminal conditions, inadequate response to curative treatment, low-income family support for patients, worried about being abandoned by the paramedics, Palliative care costs cannot be covered by personal insurance. These findings corresponding to study in [2, 17] through which it was found that financial, infrastructure, knowledge, cultural and Communications challenges [2, 17]. Alike the findings from the current study found that patients who own expenses for care were more likely to accept PC than whose medical expenses covered by public or health insurance package.

Our current study found that older age was significant associated with willingness to accept PC. These findings corresponding to a study by [18] in the United States of America, which suggests that elder people are more likely to accept hospice services and have positive attitudes and ideas about them, but it's important that they are well educated about the choice and limits of hospice care. Also, aging linkage could be a valuable outlet for distributing information related to it and other end- of- life (EoL) care options among older adults.

The current results also found that medical expenses or cost of those who reported their own expenses for care were more likely to accept palliative care than whose medical expenses were public or insurance coverage. This is different from another study conducted by [19] which found that hospital costs were necessarily related to the patient's disease. However, the cost utility from fast-track was more expensive compared to the usual care; the intervention resulting from higher quality-adjusted life years for patients was significantly higher than usual care. Thus, improved fast- truck model for specialized palliative care with psychological incomes enhanced than standard care with massive growth in cost.

Additionally, a cross-sectional study show that widespread requirements of advanced cancer patients can enlighten modifying involvements or support to their particular needs so as to enhance quality of life to them [6]. Hence, cancer patients and their caregivers have to be involved in diagnosis and decisions of their treatment this will help them to cope with disease, improve condition and adjust psychological pressure. Also, given information about end- of- life (EoL) care to cancer patients at the beginning of care will respect patients' autonomy and comfort the suffering to them particularly when they involuntarily make decisions to cancer patients in life threatening conditions.

However, this study found that the degree of accepting palliative care among advanced cancer patients was high. The result indicates significant strengths of obtaining PC at the hospital; hence patients who preferred hospital services were also willing to receive palliative care at end of life (EOL). Another study suggested that hospital remains the most mutual place of death for cancer patients [20], but this through providing explicit information through different modalities so as to improve the confidence and considerate healthcare regarding End of life (EOL) [21] conversation. Our study also showed significance in accepting tracheal intubation, artificial ventilation, artificial nutrition and hydration, tube feeding, palliative care in hospital. Similarly to another study [10], which reports the same findings health care providers should address patients' expectations with advanced cancer about aggressive treatment.

The findings of this study may contribute baseline data in Northern Tanzania, for specific tasks designed to assess the willingness to accept PC and the associated factors among advanced cancer patients; thus, there is a need for efficacy, effectiveness, and long-term impact of implemented interventions and programs focused on palliative care.

## Implication of the study

Although Tanzania has made significant steps in providing affordable palliative care, there remain a number of current challenges that need to be attended. There are quiet issues concerning to policy approaches, accessibility of palliative care, specialized training and public advocacy, which are vital when delivering high-quality care. Similarly, this study's findings can help to motivate donor agencies and Non-governmental Organization (NGO) for initiating the program towards Palliative care. However, can be used as a source of baseline research information for future research on a large scale. Also, contribute baseline data in Northern Tanzania for specific studies designed to assess the willingness to accept PC and the associated factors among advanced cancer patients; thus, there is a need for efficacy, effectiveness, and long term impact of implement interventions and programs focused on palliative care. Likewise, it is crucial to develop appropriate policy planning, applicable strategies from ministry of health and population for conducting public health programs in overall country and multi-sectorial coordination for implementing palliative care in various line activities.

Also, concerning palliative care is commonly focused on advanced stages of the disease and the end of life; it is appropriate initially in the trajectory of incurable and severe illness in combined with disease-modifying treatment. Moreover, treatment decisions, needs expectations are occasionally unpredictable; it is very important when delivering services to advanced cancer patients to offer hope to them [22]. Thus, the need for palliative care is of highly needed; and interventions that are appropriate for better quality of life throughout the disease trajectory and comfort patients balance priorities. The result for this study will adequate attention to cancer patients and assist in knowing their willingness to receive palliative care.

However, these results will help to develop the guidelines to assist with palliative care among advanced cancer patients as most of the patients present with incurable conditions; hence the need of palliative care is highly required [10]. The patient's unclear prognosis should not determine the timing of palliative involvement; somewhat, it should be decided based on the patient's ability to cooperate from palliative care [22]. Furthermore, oncology staff, the palliative care team, and other cadres will benefit from this study for skills, knowledge advancement, and strategies essential for cancer patient's care. The results will also advance excellence of cancer patients' care and help them create selections and preferences towards palliative care services.

However, not all cancer programs have adequate access to palliative care services. In this sense, we believe that our study also has important implications for designing educational programs geared at providing improved palliative care. Further, the appropriate timing of intervention should be modified in each incident programs may be adapted to features of the nurses participating in this program [23].

## Strength and limitations

The findings of our study have provided the essential data on willingness of advanced cancer patients to receive palliative care treatment. These findings were valuable in comparisons with the previous study to improve the evidence on statistical results; it will be helpful for policy-maker in creating policy to develop the guidelines to assist with palliative care among cancer patients as most of the patients diagnosed at a late stage of disease and may not be curable; thus the need of palliative care is of highly needed; also oncology staff and related cadres will benefit from this study for skills and knowledge advancement, for example, the precarious conditions psychologists involved with palliative care services.

However, this study was had some limitations need to be considered when reading the findings of this study. First, this study used a convenience sampling method to obtain participants,

which is susceptible to selection bias. Also, this was a single-centred cross- sectional study; thus, the findings from this study may not be used to generalize a whole population. Further, self-efficacy should be included as the initial screening items for assessing their end-of-life care needs and their willingness to receive palliative care (PC). Notably, our study did not recruit advanced cancer patients from the community of whom they are receiving home palliative care so it is suggestive for future study to focus in it.

## Conclusion

In the current study, willingness to accept palliative care services was more than half of the patients just 60.6% among patients with advanced cancer from the study site. The major significant determinants for willingness to receive palliative care in this study includes having better knowledge and perceived benefits of palliative care. Other factors such as advanced age of a patient, occupations, better knowledge and perceived benefits for palliative care influence to patients' willingness to accept palliative care but were lacking statistically significant. There are need for palliative care is of highly needed; and interventions that are appropriate for better quality of life throughout the disease trajectory and comfort patients balance priorities. The results for this study will adequate attention to cancer patients and assist in knowing their willingness to receive palliative care.

## Acknowledgments

The authors would like to thank all the study participants for their consents, bestowing their valuable and precious time for providing important information during data collection. We also gratefully acknowledge the assistance from editor in preparation of manuscript.

We would also like to appreciate Kilimanjaro Christian Medical Centre (KCMC) management for permitting us to conduct this study at their facility.

Our gratitude appreciation to Tertula Qulla, Neema Marandu, and Clara Kifaru for their assistances in data collection.

Also, our appreciation to Pius Data Service Centre (PDSC) for their timely and effective data entry management and quality checks.

## Author Contributions

**Conceptualization:** Grace Leonard Mushi.

**Data curation:** Grace Leonard Mushi, Julius Pius Alloyce, Vivian Frank Saria.

**Formal analysis:** Julius Pius Alloyce, Qinqin Cheng.

**Funding acquisition:** Grace Leonard Mushi.

**Investigation:** Grace Leonard Mushi.

**Methodology:** Grace Leonard Mushi.

**Project administration:** Grace Leonard Mushi, Julius Pius Alloyce.

**Resources:** Grace Leonard Mushi.

**Software:** Julius Pius Alloyce.

**Supervision:** Yongyi Chen.

**Validation:** Julius Pius Alloyce, Qinqin Cheng.

**Visualization:** Julius Pius Alloyce.

**Writing – original draft:** Grace Leonard Mushi.

**Writing – review & editing:** Furaha Serventi, Julius Pius Alloyce, Vivian Frank Saria, Xianghua Xu, Khalid Khan, Qinqin Cheng, Yongyi Chen.

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
