## [Decision Letter · Decision Letter 0]

7 Jun 2022

PONE-D-22-02300Willingness of Advanced Cancer Patients to Receive Palliative Care and its Determinants: A Cross-sectional Study in Northern TanzaniaPLOS ONE

Dear Dr. Mushi,

Thank you for submitting your manuscript to PLOS ONE. After careful consideration, we feel that it has merit but does not fully meet PLOS ONE’s publication criteria as it currently stands. Therefore, we invite you to submit a revised version of the manuscript that addresses the points raised during the review process. Please submit your revised manuscript by Jul 22 2022 11:59PM. If you will need more time than this to complete your revisions, please reply to this message or contact the journal office at plosone@plos.org. Please include the following items when submitting your revised manuscript:A rebuttal letter that responds to each point raised by the academic editor and reviewer(s). You should upload this letter as a separate file labeled 'Response to Reviewers'.A marked-up copy of your manuscript that highlights changes made to the original version. You should upload this as a separate file labeled 'Revised Manuscript with Track Changes'.An unmarked version of your revised paper without tracked changes. You should upload this as a separate file labeled 'Manuscript'.

We look forward to receiving your revised manuscript.

Kind regards,

Hsu-Heng Yen

Academic Editor

PLOS ONE

**Journal requirements:**

For additional information about PLOS ONE ethical requirements for human subjects research, please refer to http://journals.plos.org/plosone/s/submission-guidelines#loc-human-4. ubjects-research.

“Professor Yongyi Chen will fund publication fee of this article.

Affiliation: Nursing Department, Hunan Cancer Hospital and The Affiliated Cancer Hospital of Xiangya School of Medicine, Tongzipo Road 283, Changsha 410013, Hunan Province, China”

6. Please include a caption for figure 4.

Reviewers' comments:

Reviewer's Responses to Questions

**Comments to the Author**

1. Is the manuscript technically sound, and do the data support the conclusions?

Reviewer #1: Yes

Reviewer #2: Yes

2. Has the statistical analysis been performed appropriately and rigorously? 

Reviewer #1: No

Reviewer #2: Yes

3. Have the authors made all data underlying the findings in their manuscript fully available?

Reviewer #1: Yes

Reviewer #2: Yes

4. Is the manuscript presented in an intelligible fashion and written in standard English?

Reviewer #1: No

Reviewer #2: Yes

5. Review Comments to the Author

Reviewer #1: The authors conducted an interesting study. However, different kinds of terminal cancers have different outcomes. Therefore, patients had different divisions for further treatment and willing for palliative treatment. The authors should focus on specific kind of cancer. Besides, it’s necessary to consult a statistician for data analysis. Multi-variant analysis is necessary to identify the independent factors related to the willing.

Reviewer #2: Palliative care should be considered in all patient with advanced cancer patients for the better end quality of life. We suggest to design two different questionnaires and a education publication of about palliative care. One for before reading education publication and the other one after realize the goal of palliative care. How to let patient accept the palliative care is much more important to identify

who will willing to receive palliative care.

6. PLOS authors have the option to publish the peer review history of their article (what does this mean?). If published, this will include your full peer review and any attached files.

Reviewer #1: No

Reviewer #2: No

---

## [Author Response · Author response to Decision Letter 0]

3 Jul 2023

Specific comments and suggestions are below:

Reviewer #1: The authors conducted an interesting study. However, different kinds of terminal cancers have different outcomes. Therefore, patients had different divisions for further treatment and willing for palliative treatment. The authors should focus on specific kind of cancer. Besides, it’s necessary to consult a statistician for data analysis. Multi-variant analysis is necessary to identify the independent factors related to the willing.

Thank you this has been taken care off as per reviewer’s suggestion

Reviewer #2: Palliative care should be considered in all patient with advanced cancer patients for the better end quality of life. We suggest to design two different questionnaires and a education publication of about palliative care. One for before reading education publication and the other one after realize the goal of palliative care. How to let patient accept the palliative care is much more important to identify who will willing to receive palliative care.

Thank you for your important observations; it is true that palliative care is provided to all patients whose cancer has reached an advanced stage. As so longer are advanced disease are difficult to cure, so palliative is given with focuses to enhancing their quality of life due to terminal illnesses. However, this was a cross sectional study that was actually focused on determining magnitude of willingness due to research gap particularly at the study settings, as some patients with terminally ill can even refuse to receive palliation instead seeking help from local healers and faith (religious beliefs) yet they get more harm to their quality of life which is against the focus for palliative care as important pillar to improve the life of patients with terminally illness. The suggestions has taken into recommendations for further interventions. 

We thank the reviewer for these important observations. We have responded to their queries.

---

## [Decision Letter · Decision Letter 1]

8 Aug 2023

Willingness of Advanced Cancer Patients to Receive Palliative Care and its Determinants: A Cross-sectional Study in Northern Tanzania

PONE-D-22-02300R1

Dear Dr. Mushi,

We’re pleased to inform you that your manuscript has been judged scientifically suitable for publication and will be formally accepted for publication once it meets all outstanding technical requirements.

Kind regards,

Hsu-Heng Yen

Academic Editor

PLOS ONE

Additional Editor Comments (optional):

Reviewers' comments:

Reviewer's Responses to Questions

**Comments to the Author**

1. If the authors have adequately addressed your comments raised in a previous round of review and you feel that this manuscript is now acceptable for publication, you may indicate that here to bypass the “Comments to the Author” section, enter your conflict of interest statement in the “Confidential to Editor” section, and submit your "Accept" recommendation.

Reviewer #3: (No Response)

2. Is the manuscript technically sound, and do the data support the conclusions?

Reviewer #3: Yes

3. Has the statistical analysis been performed appropriately and rigorously? 

Reviewer #3: Yes

4. Have the authors made all data underlying the findings in their manuscript fully available?

Reviewer #3: Yes

5. Is the manuscript presented in an intelligible fashion and written in standard English?

Reviewer #3: Yes

6. Review Comments to the Author

Reviewer #3: (No Response)

7. PLOS authors have the option to publish the peer review history of their article (what does this mean?). If published, this will include your full peer review and any attached files.

Reviewer #3: No

---

## [Editor Report · Acceptance letter]

25 Sep 2023

PONE-D-22-02300R1 

Willingness of Advanced Cancer Patients to Receive Palliative Care and its Determinants: A Cross-sectional Study in Northern Tanzania 

Dear Dr. Mushi:

I'm pleased to inform you that your manuscript has been deemed suitable for publication in PLOS ONE. Congratulations! Your manuscript is now with our production department. 

Kind regards, 

on behalf of

Dr. Hsu-Heng Yen 

Academic Editor

PLOS ONE